# Tethered balloon-borne measurements to characterise the evolution of the Arctic atmospheric boundary layer at Station Nord

Henning Dorff<sup>1,2</sup>, Holger Siebert<sup>3</sup>, Komal Navale<sup>3</sup>, André Ehrlich<sup>1</sup>, Joshua Müller<sup>1</sup>, Michael Schäfer<sup>1</sup>, Fan Wu<sup>1</sup>, and Manfred Wendisch<sup>1</sup>

**Correspondence:** Henning Dorff (henning.dorff@uni-leipzig.de)

Abstract. We present a comprehensive balloon-borne measurement dataset collected during a dedicated Arctic observation campaign conducted from 19 March to 18 April 2024 in the transition from polar night to polar day at the Villum Research Station (Station Nord, STN, Greenland). The objective of the observations was to characterise the temporal evolution of the Arctic atmospheric boundary layer (ABL), focusing on key transition periods, including cloud development, low-level jet evolution, and day to night shifts. Data were collected by the Balloon-bornE moduLar Utility for profilinG the lower Atmosphere (BELUGA) tethered-balloon system performing in-situ measurements of temperature, humidity, wind speed, turbulence, and thermal infrared irradiance from the surface to several hundred meters altitude, with frequent profiling in high vertical resolution. Twenty-eight research flights delivered more than 300 profiles, with up to 8 profiles per hour, complemented by daily radiosonde launches. This paper specifies the BELUGA instrumentation at STN, data processing procedures, and the publicly available Level-2 data (BELUGA and radiosonde), provided in instrument-separated data subsets listed in a data collection (https://doi.pangaea.de/10.1594/PANGAEA.986431). One major application of the data is to evaluate different model types (such as numerical weather prediction, single-column, large-eddy simulations) in representing processes controlling the Arctic ABL. To prepare such evaluations, we give an overview of the observations, environmental conditions during the campaign, and highlight specific events that are valuable for model comparison. We introduce an event in which temperature rates influence the ABL inversion, radiative heating-rate profiles associated with transitions between cloudy and cloud-free conditions, and an observed Arctic low-level jet compared with reanalysis, offering insights into the Arctic ABL evolution.

## 1 Introduction

The Arctic climate system has experienced substantial changes in the past decades, including a dramatic increase in near-surface air temperature (Rantanen et al., 2022). This warming is four times higher compared to the global average and is widely known as Arctic amplification (Serreze and Francis, 2006). This phenomenon results from local and remote feedback mechanisms driven by global warming (Wendisch et al., 2023). The lapse rate and surface albedo feedbacks are considered the main drivers (Dai and Jenkins, 2023). However, models are not perfect in representing the complex physical processes that cause Arctic amplification. Steering processes such as turbulent and radiative fluxes and transitions between cloudy and

<sup>&</sup>lt;sup>1</sup>Leipzig Institute for Meteorology, Leipzig University, Leipzig, Germany

<sup>&</sup>lt;sup>2</sup>Meteorological Institute of University of Hamburg, University of Hamburg, Hamburg, Germany

<sup>&</sup>lt;sup>3</sup>Leibniz Institute for Tropospheric Research e.V., Leipzig, Germany

45

cloud-free states are limitedly represented in climate simulations, numerical weather predictions and also reanalyses (Vihma et al., 2014; Pithan and Mauritsen, 2014; Kay et al., 2016; Bromwich et al., 2018; Day et al., 2024).

A major problem of atmospheric models stems from the evolution of the Arctic atmospheric boundary layer (ABL), which significantly determines the lapse rate feedback. The ABL forms the shallow interface between the surface and the free atmosphere (Ding et al., 2017; Stroeve and Notz, 2018; Wendisch et al., 2019). The typical near-surface temperature inversion in the Arctic ABL promotes amplified near-surface warming and relatively muted heating in the free troposphere, causing the ABL to play a dominant role in the lapse rate feedback (Linke et al., 2023). However, the Arctic ABL is governed by complex processes that include radiative cooling, surface interaction, and advection.

Of particular relevance for these processes are clouds within the Arctic ABL (Morrison et al., 2011; Shupe et al., 2011; Jozef et al., 2024). Shupe et al. (2013) highlighted the impact of Arctic low-level clouds on the turbulent boundary layer structure, and they contribute to the surface radiative energy budget (Wendisch et al., 2019; Becker et al., 2023). However, the magnitude of this contribution is only partially understood (Griesche et al., 2024). Turner et al. (2018) and Lonardi et al. (2024) demonstrated that the vertical structure of the irradiance and radiative cooling rate profiles depends on the atmospheric state, especially the presence or absence of clouds. In a cloudfree atmosphere, thermal near-infrared (TIR) irradiance emission from the surface immediately results in cooling that promotes the formation of a surface-based temperature inversion. Furthermore, the presence of low-level jets (LLJs), characterised by a local maximum in the wind speed profile below 1.5 km (Tuononen et al., 2015), can modulate the Arctic ABL properties through mixing induced by enhanced wind shear and subsequent turbulence below the LLJ core (Egerer et al., 2023) and interactions with low-level cloud conditions (Neggers et al., 2025). All of these processes can lead to rapid transitions between states of the Arctic ABL with respect to atmospheric stability, turbulence, and cloud properties (Brooks et al., 2017), and become predominant in the evolution of near-surface inversions.

Understanding the temporal evolution of the transitions between different Arctic ABL states is crucial for grasping changes in the surface energy budget, cloud formation, and the evolution of Arctic weather patterns (Tjernström and Graversen, 2009; Morrison et al., 2011). Transitions can be driven locally or by advection from different regions or altitudes through air mass exchange, or along air mass trajectories (Pithan et al., 2018). Despite extensive discussions on atmospheric states (e.g., Shupe et al., 2011; Morrison et al., 2011; Pithan and Mauritsen, 2014), processes governing transitions between them, and the adjustment of the ABL are not fully understood. Numerical weather predictions, reanalyses, and climate simulations continue to face challenges in accurately representing relevant ABL transition processes (Vihma et al., 2014; Kay et al., 2016; Bromwich et al., 2018). To resolve ABL characteristics and their evolution — regarding near-surface heat fluxes, lapse rate feedback, and cloud properties — Birch et al. (2009) emphasized the importance of in situ data for all components of the surface energy budget.

Tethered balloon measurements are a suitable tool for quantifying the characteristics and evolution of the lower ABL and can bridge the gap between continuous observations from surface stations (e.g., Sedlar and Shupe, 2014) and measurements by research aircraft, which often have limited vertical resolution (Tetzlaff et al., 2015; Chechin et al., 2023). Over the past decade, balloon-borne systems have been successfully deployed in the Arctic (e.g., Sikand et al., 2013), capable of carrying payloads of up to several tens of kilograms to altitudes of 1 km, with endurances of up to several hours. From vertical profiles of measured

60

irradiances, heating rates in the Arctic ABL can be derived (Duda et al., 1991; Becker et al., 2020; Lonardi et al., 2024). Canut et al. (2016) proved the feasibility of estimating turbulence from balloon-borne 3D wind measurements.

The Balloon-bornE moduLar Utility for profilinG the lower Atmosphere (BELUGA; Egerer et al., 2019) was designed to combine measurements of thermodynamic (Egerer et al., 2021; Pilz et al., 2023), radiative (Lonardi et al., 2022; Lonardi et al., 2024), turbulent (Egerer et al., 2021; Egerer et al., 2023), and aerosol properties (Pilz et al., 2022). With slow ascent rates, BELUGA can measure sharp gradients in Arctic near-surface inversions, thereby advancing our understanding of radiative and turbulent processes that sustain and modulate the ABL inversion and cloud formation. Recent BELUGA deployments (e.g. Pilz et al., 2023; Lonardi et al., 2024) have focused on characterising different atmospheric states of the Arctic ABL.

To study specific transition phases between different ABL states, BELUGA was adapted to a lightweight and reduced payload, facilitating continuous profiling of the ABL. BELUGA was deployed in Northeastern Greenland at the Villum Research Station, Station Nord (STN), for a 1-month period (19 March to 18 April) within the framework of the Transregional Collaborative Research Center's  $\mathcal{A}rcti\mathcal{C}$   $\mathcal{A}mplification$ :  $\mathcal{C}limate$  Relevant  $\mathcal{A}tmospheric$  and Surfa $\mathcal{C}e$  Processes and Feedback Mechanisms ( $\mathcal{AC}$ )<sup>3</sup> (Wendisch et al., 2023). This deployment was in parallel to the Vertical properties of aerosols in the Arctic lower atmosphere and their impact on cloud-radiative effects (VAERTICAL)/ CleanCloud campaigns (Zamora et al., 2025). During the comprehensive measurement period, BELUGA performed intensive profiling of the ABL and captured multiple transitions between different atmospheric states. These transition phases were sampled with high temporal resolution.

This paper presents the dataset collected by BELUGA at STN and demonstrates potential applications. Section 2 introduces the STN measurement campaign, the BELUGA instrument setup and summarises the conducted research flights. Section 3 describes the data processing procedures, specifies the BELUGA Level-2 datasets and supplementary data from daily radiosonde profiles, and provides information on their accessibility. Section 4 provides an overview of the environmental conditions during the campaign that characterise the measurement data at STN. To illustrate the potential of the BELUGA measurement data, Sect. 5 highlights exemplary scientific applications for analysing Arctic ABL transition events.

#### 80 2 Tethered-balloon measurements at Station Nord

# 2.1 Station Nord

Measurements with the BELUGA system were conducted at the Villum Research Station, Station Nord (STN), from 24 March to 12 April 2024. STN is located on the northeastern coast of Greenland at 81°36'9"N, 16°40'12"W, close to a sea-ice-covered fjord in the vicinity of the Flade Isblink glacier (Fig. 1). STN is one of the northernmost manned stations in the world. It is equipped with an automatic weather station, cloud radar, lidar, and ceilometer measurements (Gryning et al., 2020). Furthermore, the parallel (VAERTICAL)/ CleanCloud campaign, conducted by the EPFL Lausanne, provided complementary aerosol data from the lower ABL using a Helikite system (Pohorsky et al., 2024).

The tethered balloon-borne measurements were operated next to the air laboratory Flyger's Hut, in a distance of roughly 2 km from the base of STN. The surrounding of Flyger's hut is mostly flat and snow-covered, elevated glaciers are located further to the south (Fig. 1b), and permanent inland ice is sufficiently distant (>100 km southwest of STN). However, the meteorological


**Figure 1.** Location of Station Nord on the northeastern tip of Greenland (purple cross), with the extent of the CARRA reanalysis (western) domain indicated by the orange rectangle (a). The mean sea ice concentration from ASMR-2 (Spreen et al., 2008), averaged over the BELUGA measurement period from 24 March to 12 April 2024, is also shown. Panel b) provides a zoomed view of the coastal region around Station Nord. Dots illustrate the grid spacing of the common-used reanalysis datasets ERA5 and CARRA, which can be evaluated using the tethered-balloon observations. The background map was created with Natural Earth.

conditions at STN are frequently influenced by katabatic flows induced by the complex orography in the hinterland of STN (e.g. Nguyen et al., 2016; Kamp et al., 2018). The location of STN is within the grid domain of the Copernicus Arctic Regional Reanalysis (CARRA), which provides a high horizontal resolution of 2.5 km for characterising local Arctic effects. CARRA adequately incorporates complex orography and sea ice conditions in the vicinity of STN (Fig. 1b).

## 5 2.2 BELUGA setup: balloon and instrument payload

The BELUGA system consists of a helium-filled tethered balloon that carries a set of dedicated in-situ measurement devices. Different balloons and sensors have been used in the past. For the STN measurement campaign (19 March to 18 April 2024), a 9 m³ balloon was used (Fig. 2a). The BELUGA instrument payload was configured to measure thermodynamic conditions, terrestrial broadband irradiance, and three-dimensional (3D) wind vectors, and was equipped with two instrument probes (summarised in Tab. 1) that rely on three of the previous probes described by Pilz et al. (2023). The two main probes are: i) the turbulence meteorological probe (TMP), which contains a hot-wire anemometer for turbulence measurements at different scales, and a modified radiosonde complemented by an ultrasonic anemometer to measure standard meteorological quantities (temperature, relative humidity, pressure) as well as the 3D wind vector (Fig. 2b), and ii) the broadband radiation probe (BP, Fig. 2c), which measures the broadband thermal infrared up- and downward irradiances. Since these probes are similar to the instrumentation detailed in Egerer et al. (2019), Lonardi et al. (2022), and Pilz et al. (2023), here we only summarize their characteristics during STN.


Figure 2. Tethered balloon at STN and instrumentation. (a) Photo of the tethered balloon system (BELUGA) at Station Nord, here fixed before launch. (b) The turbulent meteorological probe, consisting of a modified radiosonde ( $TMP_{met}$ ) and a hot-wire anemometer used for turbulence measurements ( $TMP_{turb}$ ), (c) The broadband radiometer package, which includes nadir- and zenith-pointing pyrgeometers.

Table 1. Instrument specifications of the probes deployed at BELUGA during the campaign at Station Nord.

| Probe | Instrument                  | Measured Quantities                                | Uncertainty                                 |
|-------|-----------------------------|----------------------------------------------------|---------------------------------------------|
|       | Radiosonde                  | Air temperature (T), relative humidity (RH)        | 0.2 K, 3 %-5 %                              |
| TMP   | Mini-ultrasonic anemomenter | 3D wind vector                                     | $7.5\mathrm{cm}\mathrm{s}^{-1},1.5^{\circ}$ |
|       | Hot-wire anemometer         | Wind speed at 250 Hz                               | $

**Figure 3.** Schematic sketch of typical flight strategy with instrument probes along the robe recording data in several profiles throughout the ABL. Shadings qualitatively show the temperature within the ABL with increased values (orange) around a near-surface inversion above which clouds commonly form.

This BELUGA payload is designed to minimize instrument weight while ensuring stable horizontal sensor alignment during operation. The system was also equipped with GPS and attitude sensors to account for any horizontal misalignment of the probes. Furthermore, a similar ground-based radiation probe, additionally equipped with an upward-pointing pyranometer, was operated simultaneously to measure downward solar radiative flux densities in conjunction with the BELUGA observations.

## 2.3 Flight strategy and research flights


The strategy of the measurement flights at STN aimed to characterise the vertical structure of the ABL, including near-surface inversions where low-level clouds can occur, with a temporal resolution sufficient to capture the evolution of the atmospheric parameters during transition events. The lightweight payload of BELUGA allows such high-resolution profiling of the ABL. Consequently, most flights involved continuous ascent and descent profiling, as illustrated in Fig. 3. The maximum altitude of the profiles was adjusted to the actual conditions. To gain an overview of the ABL, higher profiles up to at least 500 m height were implemented at the beginning of the flights.

In total, BELUGA conducted 28 research flights (RFs) during the STN campaign in spring 2024. All flights are listed in Table 2, which includes their respective duration, instrument status, number and maximum altitude of profiles, and atmospheric transitions observed during flight. The 28 RFs of BELUGA accumulated measurement data over about 76 flight hours, with individual flights lasting between 1 and 6 hours. The aforementioned flight strategy resulted in a large number of 336 profiles (ascent or descent) with a depth of more than 200 m. Most flights included more than ten profiles, with a maximum of 34 profiles during RF26. Profiling of the ABL was performed quasi-continuously, with ascent and descent cycles at a deca-minute frequency, to optimally capture both large-scale ABL structures and small-scale temporal fluctuations. The vertical movement of the balloon along the profiles corresponds to a mean vertical resolution of about 0.8 m, based on the averaged 1-Hz data.

**Table 2.** BELUGA research flights (RFs) with specified start and end times, data quality, and profile information (number of profiles >200 m, maximum peak height). The RFs are categorised with respect to ABL transition types that we specify in Sect. 5.

| RF   | Date       | Start    | End      | $TMP_{met}$   | $TMP_{turb}$ | BP data     | No. of   | Max    | Transition types |
|------|------------|----------|----------|---------------|--------------|-------------|----------|--------|------------------|
|      |            | time     | time     | data          | data         |             | profiles | height |                  |
| RF01 | 2024-03-24 | 12:32:56 | 14:38:13 | good          | unavailable  | unavailable | 8        | 378 m  | -                |
| RF02 | 2024-03-24 | 15:33:41 | 16:41:32 | good          | unavailable  | unavailable | 4        | 450 m  | -                |
| RF03 | 2024-03-25 | 10:09:08 | 13:56:33 | good          | unavailable  | good        | 18       | 487 m  | -                |
| RF04 | 2024-03-25 | 15:05:26 | 16:25:43 | unavailable   | unavailable  | good        | 6        | 406 m  | -                |
| RF05 | 2024-03-26 | 16:07:39 | 18:25:35 | good          | unavailable  | good        | 12       | 482 m  | night⇔day        |
| RF06 | 2024-03-26 | 19:09:02 | 21:57:37 | good          | unavailable  | good        | 12       | 465 m  | night⇔day        |
| RF07 | 2024-03-27 | 16:11:40 | 17:42:30 | good          | unavailable  | good        | 6        | 480 m  | cloudfree↔cloudy |
| RF08 | 2024-03-28 | 09:56:12 | 12:48:06 | good          | unavailable  | good        | 12       | 494 m  | cloudfree⇔cloudy |
| RF09 | 2024-03-28 | 13:41:48 | 17:02:12 | unavailable   | unavailable  | unavailable | -        | -      | -                |
| RF10 | 2024-03-29 | 14:41:38 | 16:05:56 | good          | unavailable  | good        | 4        | 474 m  | cloudfree⇔cloudy |
| RF11 | 2024-03-30 | 10:45:28 | 14:09:54 | unavailable   | unavailable  | good        | 10       | 489 m  | cloudfree⇔cloudy |
| RF12 | 2024-04-01 | 09:35:37 | 11:49:08 | good          | unavailable  | good        | 12       | 688 m  | LLJ              |
| RF13 | 2024-04-01 | 12:13:25 | 14:48:22 | good          | good         | good        | 12       | 549 m  | LLJ              |
| RF14 | 2024-04-01 | 15:17:39 | 16:59:37 | good          | good         | good        | 6        | 502 m  | -                |
| RF15 | 2024-04-02 | 09:31:52 | 12:35:30 | good          | good         | good        | 20       | 475 m  | cloudfree⇔cloudy |
| RF16 | 2024-04-02 | 13:08:47 | 14:27:02 | good          | good         | good        | 6        | 543 m  | cloudfree⇔cloudy |
| RF17 | 2024-04-03 | 13:31:30 | 16:17:28 | good          | good         | good        | 10       | 616 m  | -                |
| RF18 | 2024-04-04 | 10:47:22 | 13:47:51 | good          | good         | good        | 20       | 507 m  | -                |
| RF19 | 2024-04-04 | 14:25:02 | 17:07:03 | good          | unavailable  | good        | 20       | 549 m  | cloudfree⇔cloudy |
| RF20 | 2024-04-05 | 09:36:28 | 14:59:31 | good          | good         | unavailable | 24       | 599 m  | -                |
| RF21 | 2024-04-06 | 14:44:10 | 17:12:06 | good          | good         | good        | 16       | 572 m  | -                |
| RF22 | 2024-04-07 | 12:40:48 | 15:52:54 | good          | good         | good        | 18       | 717 m  | -                |
| RF23 | 2024-04-08 | 09:33:23 | 12:48:46 | good          | good         | good        | 14       | 734 m  | cloudfree↔cloudy |
| RF24 | 2024-04-08 | 13:53:34 | 16:26:09 | good          | good         | good        | 12       | 769 m  | cloudfree⇔cloudy |
| RF25 | 2024-04-09 | 09:12:10 | 13:36:16 | missing parts | unavailable  | good        | 25       | 855 m  | -                |
| RF26 | 2024-04-10 | 09:56:37 | 15:41:44 | good          | good         | good        | 34       | 829 m  | LLJ              |
| RF27 | 2024-04-11 | 09:38:26 | 14:08:32 | gaps          | good         | good        | 24       | 897 m  | LLJ              |
| RF28 | 2024-04-12 | 13:58:35 | 16:29:16 | unavailable   | unavailable  | good        | 16       | 819 m  | -                |

The maximum altitude attainable for BELUGA is determined by wind conditions, as stronger winds increase horizontal drift, requiring a longer tether that adds weight and consequently reduces the achievable height. For the STN campaign, Fig. 4 shows the maximum heights of all profiles above 200 m, with each RF depicted individually. Most profiles exceeded 400 m, with the highest flight altitude reaching nearly 900 m during RF27. Beyond wind requirements, the flight strategy was slightly adapted throughout the campaign. In the first half of RFs, maximum profile heights were generally set at similar altitudes. In the second half, profile heights varied more, also within individual RFs, and frequently exceeded 500 m. The change was motivated by higher variability of near-surface temperature inversions in the second half, which also resulted in a higher number of profiles.

**Figure 4.** Maximum heights of profiles for each research flight (RF) of the tethered-balloon. Markers are colour-coded with respect to the ABL transition types during which the flight were performed.

#### 3 Level-2 data



For the measurements of the devices described in Sect. 2.2, we published the post-processed Level-2 (L2) data at the World Data Center PANGAEA (Felden et al., 2023), under an open access licence (CC-BY 4.0). The L2 measurements are compiled in a data collection titled *Balloon-borne profile L2-data characterising the Arctic boundary layer and troposphere at Station Nord* (Dorff et al., 2025c), accessible via the link https://doi.pangaea.de/10.1594/PANGAEA.986431. Table 3 summarises the access to the individual instrument probe and radiosonde datasets. For each probe and research flight, single Network Common Data Format (netCDF) files (version netCDF-4) are provided, referencing UTC time. The file naming convention includes the measurement platform (BELUGA or Radiosonde), the instrument probe type (TMP<sub>met</sub>, TMP<sub>turb</sub>, BP), the RF label (RFxy) and the date. Following the Climate and Forecast (CF) Metadata Conventions version 1.12 (Eaton et al., 2024), global attributes provide general data information, while variable attributes specify units and long names for each variable. Here, we describe the common altitude reference, the segmentation of flight sections, data quality control procedures, and give an overview of the post-calibrated L2 observations.

#### 3.1 Common altitude reference

The L2-data for each BELUGA probe component are referenced to the barometric altitude in metres, as all probes are equipped with appropriate sensors measuring static barometric pressure  $p_b$ . Barometric values used as altitude reference are more reliable in the high Arctic than GPS-based values. From  $p_b$ , we calculate the barometric altitude  $z_b$  in metres as follows:

$$z_{\rm b} = \frac{T_0}{L_0} \cdot \left( 1 - \frac{p_{\rm b}}{p_0} \frac{\frac{R \cdot L_0}{g}}{p_0} \right) \tag{1}$$

using the standard adiabatic lapse rate  $L_0$ =6.5 K km<sup>-1</sup> and the specific gas constant for dry air  $R_L$ =287 J kg K<sup>-1</sup> (Wendisch and Brenguier, 2013). To correct for synoptic pressure tendencies and to ensure comparability in barometric altitude estimates,

Table 3. Specifications of published L2 BELUGA data sets from Station Nord.

| Probe                            | Provided quantities                                                            | Reference                                                           |
|----------------------------------|--------------------------------------------------------------------------------|---------------------------------------------------------------------|
| $TMP_{met}$                      | barometric pressure, air temperature, relative humid-                          | Dorff et al. (2025f): https://doi.pangaea.de/10.1594/PANGAEA.987003 |
|                                  | ity, hor. wind speed and direction, lateral and longitu-                       |                                                                     |
|                                  | dinal wind vector components, barometric altitude                              |                                                                     |
| $\overline{\mathrm{TMP_{turb}}}$ | static pressure, temperature, raw hot-wire signal, air                         | Dorff et al. (2025e): https://doi.pangaea.de/10.1594/PANGAEA.987008 |
|                                  | density, calibrated wind speed and barometric altitude                         |                                                                     |
| BP                               | downward and upward terrestrial irradiances ( $F^{\downarrow}$ and             | Dorff et al. (2025b): https://doi.pangaea.de/10.1594/PANGAEA.987001 |
|                                  | $F^{\uparrow}$ ), net terrestrial irradiance $F_{ m net}$ , air pressure, tem- |                                                                     |
|                                  | perature and relative humidity, and barometric altitude                        |                                                                     |
| Radiosonde                       | air pressure, air temperature, capacitor temperature,                          | Dorff et al. (2025a): https://doi.pangaea.de/10.1594/PANGAEA.987053 |
|                                  | relative humidity, speed of the sonde, u- and v wind,                          |                                                                     |
|                                  | distance, altitude                                                             |                                                                     |

surface pressure  $p_0$  and temperature  $T_0$ , along with their tendencies during each flight were obtained from a ground-based pyrgeometer equipped with a BME-sensor. These measurements were time-synchronised with the balloon data.

## 3.2 Flight segmentation


The BELUGA datasets listed in Table 3 are provided with flight segmentation. The segmentation is stored as a one-dimensional, time-dependent flag that specifies distinct periods of flight segments. Data from specific segments can be easily accessed via the data variable <code>flight\_segments</code> and individual IDs, e.g. <code>profile\_ascent\_01</code>. For RF01, Fig. 5 illustrates the categories of flight segments identified by our segmentation. Profiles are flagged for ascents and descents when vertical movement rates of BELUGA exceed  $0.3 \, \mathrm{m \, s^{-1}}$  (averaged over 30 s). More than 80 % of the flight duration is assigned to profiles due to the observational focus on vertical conditions in the Arctic ABL. Additionally, the data include periods where BELUGA maintained a relatively constant altitude for several minutes (Fig. 5; green). These periods are identified by mean vertical movement rates below  $0.3 \, \mathrm{m \, s^{-1}}$  when averaged over 30 s. Such height-constant segments provide a statistical basis for turbulent flux estimates and help characterise the temporal evolution of radiative cloud properties at a given altitude. Measurement periods near the surface, such as at the beginning of RF01 (Fig. 5), result from the installation of the probes along the balloon tether, or, in other cases, aim to compare the balloon-borne instruments with ground-based near-surface measurements from STN.

#### 3.3 Quality control

The L2-data have been quality-checked and appropriately flagged. Outliers were removed, and spike filtering was performed for each quantity. For meteorological quantities, physically implausible values were identified. Specifically, temperature values above 0°C were set to NaN, as were unrealistic wind speeds exceeding 20 m s<sup>-1</sup>. Values of RH were capped at 120 %. In the data quality flag, these cases were labeled as *missing* and *bad* accordingly. Sensor tilt of the BP can significantly affect the

**Figure 5.** Flight altitude over time during flight (RF01) with all flagged flight segments. Vertical profiling is distinguished by ascent and descent. Peaks are flagged for 30 s before and after maximum profile heights (red dots, with the highest profile depicted as darked dot).

accuracy of irradiance values; data for sensor pitch angles  $> 15^{\circ}$  were treated as *bad*. Data gaps up to 10 s were interpolated and flagged accordingly. For available data, the quality flag assigns *good* quality for almost 95 % of all flight periods.

#### 3.4 Level-2 data overview




# 3.4.1 Meteorological data from the TMP

The meteorological L2-data for the  $TMP_{\rm met}$  (Dorff et al., 2025f, https://doi.pangaea.de/10.1594/PANGAEA.987003) are provided at 1 Hz resolution and contain the following quantities: barometric pressure, air temperature (radiosonde and sonic), relative humidity, horizontal wind speed, wind direction, longitudinal and lateral wind vector components and barometric altitude (Table 3). The three attitude angles (roll, pitch, and yaw) from the navigation module were used to transform the 3D wind vector into a ground-fixed coordinate system. However, the design of the sonic does not allow for a precise determination of the vertical wind component.

Figure 6 summarises the distribution of meteorological values from the  $TMP_{met}$  data as a function of height, which were observed throughout the measurement period. The distributions correspond to a total measurement duration of more than  $10\,h$  for profile data at the lowest levels, decreasing with height due to varying profile heights; however, data still accumulate to about  $1\,h$  above  $500\,m$  (Fig. 6a). Temperature values are predominantly below  $-10\,^{\circ}$ C throughout the ABL (Fig. 6b) with two notable nodes at inversion heights below  $200\,m$ . Overall, quite dry conditions were observed, with RH mainly ranging between  $40\,and\,60\,\%$  in the lowest hundred metres. As altitude increases, the spread of the RH distributions widens (Fig. 6c), indicating a higher variability of moisture conditions with height and the sporadic presence of clouds. The wind was generally calm (below  $8\,m\,s^{-1}$  at all heights, and particularly below  $4\,m\,s^{-1}$  close to the surface). Between  $50\,m$  to  $200\,m$ , there is an increasing frequency of wind speeds exceeding  $4\,m\,s^{-1}$ , associated with the occurrence of low-level jets (LLJs).

Figure 6. Summary of meteorological data from the  $\mathrm{TMP}_{\mathrm{met}}$  package recorded during the measurement period. a) shows the distribution of the measurement heights and respective flight duration in each height bin. For these data, b) illustrates the statistics of measured temperature values as a function of height. c) and d) analogously depict the values of relative humidity and wind speed, respectively.

## 3.4.2 Turbulence data from the TMP


The L2-data of the turbulence component (TMP<sub>turb</sub>; Dorff et al., 2025e, https://doi.pangaea.de/10.1594/PANGAEA.987008) are provided at a temporal resolution of 50 Hz and include the following quantities: static pressure, temperature, raw hot-wire signal, air density, barometric height and calibrated wind velocity with 50 Hz resolution (Table 3). The raw data sampled at 250 Hz was averaged over 5 data points to reduce noise levels. The hot-wire anemometer wind speed was calibrated individually for each flight using the horizontal wind measurements from the sonic anemometer. For calibration, we applied a least-squares method similar to that described in Frehlich et al. (2003), which accounts for dependencies on air temperature and density.

Static pressure, air temperature and density values are included in the TMP<sub>turb</sub> data. For consistency, these were upsampled from initially 1 Hz to 50 Hz.

The calibrated wind speed values enable the resolution of turbulence at small scales through the calculation of power spectra of wind velocities. Figure 7 shows the power spectral density of the high-resolution wind speed derived from an exemplary 20-minute measurement period conducted at a constant height of 10 m. The raw spectrum was additionally averaged over logarithmically equidistant bins, and referenced to a model spectrum for an inertial range with a slope of -5/3. Even in the high frequency range, the spectrum in Fig. 7 does not yet transition to white noise (a flat spectrum), implying that spatial structures of about 2 cm can still be well resolved. Since the spectrum is normalised such that its integral yields the variance, the sensor resolution can be estimated as  $dU = \sqrt{f_{\rm n} \cdot 50\,{\rm Hz}} \approx 7\,{\rm mm\,s^{-1}}$  for the noise level of the spectrum  $f_{\rm n} = 10^{-6}{\rm m}^{-2}\,{\rm s}^{-2}\,{\rm Hz}^{-1}$ .

**Figure 7.** Power spectral density of calibrated hot-wire wind data measured during a 20 min sampling period at approximately 10 m above the surface during RF22 (7 April 2024). The raw spectrum (blue), the spectrum averaged over logarithmically equidistant bins (orange), and a slope of -5/3 for the inertia subrange (green) are shown.

# 3.4.3 Broadband radiation probe (BP)



The L2-data of the broadband radiation probe (BP; Dorff et al., 2025b, https://doi.pangaea.de/10.1594/PANGAEA.987001) are provided at a temporal resolution of 1 Hz. We calibrated the broadband irradiance measurements following the methods introduced by Egerer et al. (2019), Lonardi et al. (2022), and Pilz et al. (2023). The sensor senstivity specified in the manufacturer's certification was applied. Additionally, the effects of sensor inertia due to the finite response times of the radiometers were corrected using the method by Ehrlich and Wendisch (2015). Subsequently, a low-pass filter was applied to reduce noise, employing a moving-window average with a period of 7 s. From the calibrated downward and upward irradiances, the net TIR irradiance,  $F_{\text{net}}$ , was calculated as  $F_{\text{net}} = F^{\downarrow} - F^{\uparrow}$ , which quantifies the infrared radiative energy budget at flight altitude. Accordingly, the BP L2-dataset includes the following quantities: downward and upward terrestrial irradiances ( $F^{\downarrow}$  and  $F^{\uparrow}$ ), the net terrestrial irradiance  $F_{\text{net}}$ , measurements of air pressure, temperature, and relative humidity from the BME-280, as well as barometric altitude (Table 3).

Summarising all flights, Figure 8 displays the frequency distribution of  $F_{\rm net}$  values, comparing data measured at approximately 30 m height (near-surface) and around the profile peak heights, which range from 200 to 900 m (Fig. 4). Both frequency distributions reveal  $F_{\rm net}$  values spanning from -100 W m<sup>-2</sup> (net upward radiation) to 0 W m<sup>-2</sup>, with barely positive values (net downward radiation). Significant differences can be identified between the two distributions and heights, highlighting the relevance of vertical profiling the radiative energy budget. In particular, there is a marked negative shift in the distribution curve toward more negative  $F_{\rm net}$  values at profile peak heights. These differences between the near-surface and upper-level irradiance distributions are the result of vertical variation in cloud cover, with low-level clouds causing negative irradiances.

Figure 8. Frequency distribution of net thermal infrared irradiances ( $F_{net}$ ) for near the surface (purple) considering data around 30 m height, and for the profile peaks (orange), if the maximum height was higher than 200 m. All flight periods with BP measurements are included.

## 3.4.4 Daily radiosondes




Profile measurements by radiosondes that were launched each morning prior to the first BELUGA ascent are published (Dorff et al., 2025a, https://doi.pangaea.de/10.1594/PANGAEA.987053) to complement the BELUGA observations of ABL profiles. These radiosonde profiles cover the full troposphere and thus facilitate further analysis by providing information about the background meteorological (synoptic) state above the ABL and above the BELUGA observations. The radiosondes used were of type DFM-17 (GRAW), measuring air pressure, temperature, relative humidity, wind speed, and wind direction. The upper-tropospheric profiles, especially regarding relative humidity are particularly relevant for locating clouds and moist areas above the ABL. Consequently, these profiles contributed to the identification of clear-to-cloudy transition events. The radiosonde data have been post-processed by quality checking the physical plausibility of the measurement values.

Figure 9 illustrates the meteorological standard quantities measured by all radiosonde launches during the campaign period at STN (spring 2024). For a few days (27 and 30 March, 3 April), the radiosonde data were limited to below 5 km altitude due to data transfer issues. Overall, the radiosonde profiles indicate a predominantly stable stratified and mostly dry troposphere throughout the campaign. Winds remained generally calm, but they significantly intensified in the middle of the campaign.

## 250 4 Meteorological conditions during campaign

During the measurement period, Arctic sea ice extended along nearly the entire Greenland coast (Fig. 1a), including the coastline near STN (Fig. 1b). When marine air masses influence the local conditions at STN, their thermodynamic vertical structure is likely affected by traversal over sea-ice-covered ocean. In addition, katabatic flows from the south, driven by the complex orography in the hinterland of STN, can frequently influence the synoptic conditions, which are prerequisites for the measurement data we recorded at STN.



**Figure 9.** Radiosonde profiles of (a) potential temperature, (b) relative humidity, (c) wind speed for the entire balloon-borne measurement campaign period in spring 2024 at STN. Profiles are only shown up to heights of 10 km.

We use the CARRA reanalysis data (Schyberg et al., 2020) to briefly describe the synoptic conditions. Maps of the equivalent potential temperature  $\Theta_e$  at 850 hPa and the 500 hPa geopotential heights in Fig. 10 illustrate how the synoptic patterns changed considerably over the measurement campaign period. During the first week of the measurement period (20-27 March 2024, Fig. 10a), an expansive high-pressure ridge over the North Atlantic advected rather cold and dry air masses from the pole towards the STN. Nevertheless, the flow remained weak because the geopotential gradients were relatively small. The synoptic situation changed during the second week when the northeastern part of Greenland and the Fram Strait became increasingly influenced by a low-pressure system northeast of Svalbard (Fig. 10b). Although southwestern Greenland remained under the influence of the high-pressure ridge, the geopotential gradients over STN substantially increased, favoring intensified northwestern winds and the advection of cold polar air masses. The approaching low pressure system also increased the likelihood of cloud formation at STN.

In the first week of April (Fig. 10c), this pattern persisted in general; however, the geopotential dipoles declined, reducing the pressure gradients over northeast Greenland and surface winds. The decrease in advection of polar air masses caused  $\Theta_{\rm e,850\,hPa}$  to rise to approximately 265 K at STN. In the last week of the campaign, particularly on 10 April (Fig. 10d), a significant change occurred in the synoptic pattern due to the formation of a low-pressure system over central North Greenland (with 

Figure 10. CARRA-based synoptic overview of campaign period depicting four days (2024-03-23, 2024-03-30, 2024-04-05, 2024-04-10). For each day, filled contours show equivalent potential temperature  $\Theta_e$  at 850 hPa. Isolines show geopotential metre at the 500 hPa level.

These large-scale synoptic patterns are reflected in the local conditions at STN, as illustrated over the entire measurement period using CARRA data in Fig. 11. During the first week, when weak pressure gradients prevailed, winds were calm at STN, favoring stable stratification, as indicated by the spread of  $\Theta$  between 50 m and 500 m altitude (Fig. 11a, b). High RH values near the surface highlight the potential presence of a near-surface inversion, which conserves moisture close to the ground (Fig. 11c). The approaching low pressure system in the second week led to a significant increase in wind speeds exceeding  $10 \,\mathrm{m\,s^{-1}}$  at STN (Fig. 11b). Notably, when the upper wind fields decreased at the beginning of April, a LLJ persisted with a strong


**Figure 11.** Meteorological conditions during the measurement campaign period (2024-03-24 to 2024-04-10) as represented by CARRA reanalysis data interpolated onto the measurement station location. a) shows potential temperature values for 50 m and 500 m height above ground over the measurement period together wind speed values at 50 m on the additional right y-axis. b) depicts the vertical profiles of wind speed for the lowest 500 m. c) represents the vertical profiles of relative humidity.

reduction in humidity, which was associated with significant warming of the ABL (Fig. 11a). For the rest of the campaign,  $\Theta$  values remained relatively constant; however, around April 10, the vertical profiles of wind speed indicate another LLJ event (Fig. 11b), with higher humidity located below this jet (Fig. 11c). Note that the radiosonde profiles in Fig. 9 generally confirm the synoptic development indicated by CARRA, highlighting rather dry conditions at Station Nord.

#### 5 Data potential to investigate transition events

While the stratification of the ABL, as well as moisture and wind conditions, changed considerably throughout the measurement period (Sect. 4), numerous transitions between the ABL states occurred. We classified three major categories of transition events based on visual data inspection: (1) changes in cloud conditions between clear and cloudy states, (2) LLJ evolution, and (3) day-to-night transitions. When these transitions occurred during flight, they are documented in Table 2 and Figure 11, and are also marked as attributes in the L2-data. Accordingly, fifteen RFs recorded significant transitions that modulated the Arctic ABL conditions. The following section sketches how the BELUGA L2-data from STN during such transitions can enhance our understanding of Arctic ABL steering processes.

Figure 12. Near-surface thermal structure and its temporal evolution during RF15 (2 April 2024) based on  $TMP_{met}$  data. (a) Flight altitude with color corresponding to time on the x-axis. color-coded progression over time. (b) Vertical profiles of potential temperature ( $\Theta$ ), with each profile color-matched to the corresponding time segment in (a). (c) Calculated potential temperature tendency in  $Kh^{-1}$  as vertical profile, referring to the difference between the first and last  $\Theta$  profiles in (b). Positive (negative) values indicate warming (cooling).

# 290 5.1 Temperature tendencies in near-surface inversions


Among the transitions between Arctic ABL states, changes in cloud conditions have a significant impact on the Arctic ABL structure, particularly evident in variations of the vertical temperature gradient (lapse-rate) at small spatio-temporal scales. The repetitive BELUGA profile observations, with their high temporal resolution, enable resolving such thermodynamic impacts.

Figure 12 shows exemplary vertical potential temperature profiles from 2 April 2024 (RF15), sampled under varying cloud and moisture conditions. High RH values above 5 km, measured by the radiosonde (Fig. 9), suggest the presence of mid-to high-level (cirrus) clouds, as confirmed by local ceilometer data (not shown) showing that they were descending and gradually dissipating during flight. At the start of the profiling sequence, a pronounced near-surface temperature inversion was present within the lowest 100 m. The inversion weakened in successive profiles, likely due to enhanced turbulent mixing associated with increasing surface heating. Later profiles reveal a smoother potential temperature variation with height (Fig. 12b).

320

325

For the sequence of temperature profiles, we calculated the temperature tendencies, which are, as displayed in Fig. 12c, based on first and last profiles (with a time gap of approximately 2.75 h). The tendencies indicate modest cooling above 300 m, slight warming between 200 and 300 m, and near-surface cooling or reduced heating. The tendencies are statistically significant and robust, as indicated by the calculated uncertainty ranges (Fig. 12c). We estimated the uncertainties using Gaussian error propagation, assuming a constant temperature measurement uncertainty of  $\pm 0.2$  K, as specified for the sensor in Table 1.

Taking uncertainties into account, the results demonstrate that even subtle thermal tendencies can be detected with confidence by the BELUGA observations. This example highlights the capability of BELUGA's high-resolution profiling to investigate ABL inversion thermodynamics, particularly during transitions under varying cloud conditions.

#### **5.2** Evolution of heating rates in cloudfree ← cloudy transitions

Temperature tendencies in the near-surface inversion between cloud-free and cloudy transitions, as described in Sect. 5.1, can be induced by advection (horizontal exchange of different air masses) or by the divergence of the local radiative energy budget. The divergence of radiative energy, measured by BELUGA with the BP, can be translated into radiative heating rates. The local radiative heating rate is defined as the temporal change in temperature resulting from variations in net irradiance  $\delta F_{\rm net}$  with altitude z (Egerer et al., 2019). Following Lonardi et al. (2024) and using the downward and upward terrestrial irradiance  $(F^{\downarrow}(z), F^{\uparrow}(z))$  measured by the BP at a given height z, the radiative heating rates  $\zeta$  are calculated as:

$$\zeta(z) = \frac{1}{\rho \cdot c_{\rm p}} \frac{\Delta}{\Delta z} \left( \underbrace{F^{\downarrow}(z) - F^{\uparrow}(z)}_{F_{\rm net}} \right),$$
 (2)

where  $\rho$  represents the air density and  $c_{\rm p}$  is the specific heat capacity of air at constant pressure. For the calculations, a single BELUGA profile is used, assuming that the environment remains in a steady state during vertical profiling of the ABL. A layer thickness of  $\Delta z = 10\,\mathrm{m}$  was chosen.

Reconsidering the thermal inversion under changing cloud conditions during RF15, we, similar to Sect. 5.1 and Fig. 12, analyse the first and last profile to demonstrate the capabilities of BELUGA to study heating rate profiles during a transition event. Their net irradiances and calculated heating rates are shown in Fig. 13. While the intensification of the net upward irradiance remains relatively consistent between both profiles up to  $150 \, \text{m}$ , the last profile exhibits a much stronger vertical decay of  $F_{\text{net}}$ , even though  $F_{\text{net}}$  values become less negative around the inversion bottom (Fig. 13a). The calculated heating rates (Fig. 13b) are approximately half the magnitude of the (potential) temperature tendencies identified in Fig. 12. The negative heating rates (cooling) suggest that advection somewhat overcompensates radiative effects on the lapse rate, especially near the inversion at around 200 m. This inversion height corresponds to the vertical level with the strongest cooling and heating rates. This vertical variability of radiative heating rates, as shown in Fig. 13, emphasizes the importance of quantifying thermal irradiance as a function of height to better understand the evolution of the ABL with respect to the thermodynamic and radiative effects modulating the ABL structure.



Figure 13. BELUGA observed net irradiance  $F_{\text{net}}$  as a function of height showing the first and last profile of RF15 (a). From these profiles, heating rates  $\zeta$  are calculated using Eq. 2 and their mean is derived.

#### 5.3 Low-level jet evolution compared to CARRA

Low level jets (LLJs) also play a significant role in modulating the Arctic ABL structure through entrainment. Previous balloon-borne measurements have demonstrated that LLJs can effectively enhance near-surface horizontal transport of passive tracers (Egerer et al., 2023). During the measurement campaign at STN, LLJ events were sampled according to Tab. 2 and Fig. 11. For the LLJ event on 1 April 2024 (during RF12), Fig. 14 presents the balloon-borne meteorological conditions in terms of observed vertical profiles of potential temperature, wind speed, and specific humidity. All measurements from RF12 are compared here to the corresponding CARRA reanalysis data for the duration of the flight.

The BELUGA observations indicate a pronounced LLJ with a maximum wind speed approaching 5 m s<sup>-1</sup> above a stably stratified Arctic ABL, extending up to 100 m (Fig. 14a, b). During the flight, the LLJ intensity increased by approximately  $1 \,\mathrm{m\,s^{-1}}$  (Fig. 14b). Aside from a decline in the near-surface layer (

**Figure 14.** Flight pattern of RF12 during 1 April 2024 (a), with measured vertical profiles of potential temperature (b), horizontal wind speed (c), and specific humidty (d) by BELUGA. The observations are compared to the corresponding CARRA data (bold dots) which are interpolated onto the STN location and considers the model time steps that correspond to the flight duration. All data is shown as a function of height above ground level (AGL).

vertical measurements of meteorological, radiative, and turbulence conditions facilitate a better understanding of the limitations inherent in current state-of-the-art simulations, particularly in representing transition events in Arctic ABL conditions.

## 6 Data availability


All processed BELUGA and radiosonde Level-2 data are publicly available at the World Data Center PANGAEA under CC-BY 4.0 (Table 3). The entire data set is compiled in the data collection entitled *Balloon-borne profile L2-data characterising the Arctic boundary layer and troposphere at Station Nord* (Dorff et al., 2025c; https://doi.pangaea.de/10.1594/PANGAEA.986431). All included netCDF files follow the Climate and Forecast (CF) Metadata Conventions version 1.12 (Eaton et al., 2024). Details about the raw and Level-1 measurement files can be obtained upon request.

We strongly encourage the use of balloon-borne L2-data from STN as a continuation of a series of previous BELUGA measurement campaigns with similar instrumental setups, such as during the Physical feedbacks of Arctic planetary boundary level Sea ice, Cloud and AerosoL (PASCAL) campaign (Wendisch et al., 2019; Egerer et al., 2019; Egerer et al., 2021), the





Multidisciplinary drifting Observatory for the Study of Arctic Climate (MOSAiC) expedition (Shupe et al., 2022; Lonardi et al., 2022; Pilz et al., 2023), and at Ny-Ålesund, Svalbard (Lonardi et al., 2024).

The CARRA data used to analyse the meteorological conditions are publicly available, provided by Schyberg et al. (2020). This CARRA dataset includes eleven specific height levels, ranging from 15 m up to 500 m.

The code for processing the balloon-borne measurements from Level-0 (raw) to Level-2 (published) is python-based and accessible via Zenodo (Dorff et al., 2025d). The processing code framework employs an object-oriented architecture, with the main class *BELUGA* composed of three subclasses specific to the instrument probes. These subclasses contain all routines required for the post-processing of measurements, complemented by plotting functions to visualise processing steps and data analysis.

# 7 Summary and conclusions

As part of the  $(\mathcal{AC})^3$  project, this paper presents an observation-based dataset comprising balloon-borne measurements within the Arctic boundary layer, obtained using the Balloon-bornE moduLar Utility for profilinG the lower Atmosphere (BELUGA) system. BELUGA is a tethered balloon equipped with multiple atmospheric sensors, enabling high-resolution vertical profiling of temperature, humidity, wind fields, turbulence, and broadband thermal-infrared irradiance. The dataset described herein originates from a comprehensive measurement campaign conducted at the Villum Research Station, Station Nord (STN, 81° 36′ 9″ N, 16° 40′ 12″ W), Greenland, from 19 March 2024 to 18 April 2024. The processed Level-2 measurement data are publicly available and are compiled in a PANGAEA data bibliography (Dorff et al., 2025c), organized into probe-separated subsets providing netCDF files for each individual research flight (RF).

Overall, a total of 28 BELUGA RFs were conducted during the measurement period in spring 2024 at STN. The measurement strategy aimed to perform continuous profiling throughout the Arctic atmospheric boundary layer (ABL). The RFs collected measurement data over approximately 76 flight hours and yielded 336 deep vertical profiles, sampling thermodynamic (T, RH, p), radiation (up- and downward terrestrial near-infrared irradiance;  $F^{\uparrow}$  and  $F^{\downarrow}$ ) and turbulence parameters (eddy dissipation rate  $\epsilon$ ) within the Arctic ABL. Some profiles reached altitudes of up to 900 m. During each flight, measurements were performed mainly in a continuous manner for several hours, limited only by battery constraints.

This BELUGA data from STN enable the investigation of the temporal evolution of the ABL with near-surface inversions under various atmospheric transitions, including the day-to-night transitions, cloud cover variations, and low-level jet (LLJ) formation. We presented examples of observed transitions and highlighted the scientific applicability of the published balloon-borne observations for understanding temperature and heating rates occurring in near-surface inversions, as well as the radiative energy budget during Arctic low-level cloudfree-cloudy transitions, and thermodynamic characteristics during LLJ formation.

These observations offer valuable insights into Arctic ABL transitions, emphasizing the roles of near-surface thermal inversions, cloud-radiative effects and wind structure evolution. By shedding light on these processes, BELUGA data significantly enhance the understanding and quantification of surface warming via the lapse rate feedback, which is driven by temperature lapse rate (stability), inversion height and strength, vertical humidity, air mass circulation, and cloud processes. Current climate

https://doi.org/10.5194/essd-2025-651
Preprint. Discussion started: 24 November 2025

© Author(s) 2025. CC BY 4.0 License.





Science Science Data

models still face challenges in adequately resolving the relevant small-scale processes within the ABL, although the lapse-rate feedback is recognised as the primary driver of Arctic amplification (Pithan and Mauritsen, 2014; Schneider et al., 2021).

For this reason, we encourage synergy studies that combine the BELUGA observations with various modeling frameworks to examine the ABL evolution and their impact on the lapse-rate feedback at different scales. Single column models (SCMs) can quantify the influence of clouds on ABL evolution, also for heights unreachable for the balloon. SCMs are computationally efficient and suitable for inexpensive parameterization testing and sensitivity experiments, using BELUGA measurements as input. Large-eddy simulations (LES) can elucidate horizontal cloud scenarios and resolve dynamical boundary layer processes on larger scales, but are associated with high computational costs. Comparing the observations with NWP models can assess how accurately transitions between different states of the Arctic ABL and their influence on surface warming via the lapse-rate feedback are represented in longer-term projections. Additionally, BELUGA data are crucial for evaluating reanalyses such as the ECMWF Reanalysis v5 (ERA5) and the Copernicus Arctic Regional Reanalysis (CARRA).

Together with the supplementary observational platforms installed at STN during spring 2024 (Sect. 2.1) and the aforementioned model configurations, the BELUGA L2-data are embedded within a manifold framework, which enhances the ability to simulate Arctic ABL processes under varying atmospheric conditions and to understand how their contributions to Arctic amplification via feedback mechanisms. The synergy of these datasets will provide an unprecedented and comprehensive characterisation of the Arctic ABL. This enables greater insight into the role of surface warming driven by the lapse-rate feedback, with balloon-borne observations serving as benchmarks for model-based, process-oriented sensitivity studies.

410 Author contributions. MW and HS were the main initiators for the conceptualisation of the measurement approach and its acquisition. HS and ML conducted the balloon-borne measurements at Station Nord. HD was in charge for post-processing the balloon-borne data with fundamental support for the TMP data by HS and KN, and support for the BP data by AE and FW, JM, and support for the radiosonde data by JM. The publication of the post-processed data was led by HD. HD conceptualised and drafted the manuscript and visualisations, while HS and KM were responsible for specific sections concerning turbulence measurements (HS) and scientific analysis of temperature rates 415 (KM). All authors were involved in revising the manuscript.

Competing interests. The authors declare that they have no conflict of interest.

Acknowledgements. We gratefully acknowledge the funding by the Deutsche Forschungsgemeinschaft (DFG, German Research Foundation) – project number 268020496 – within the Transregional Collaborative Research Center TRR 172, "ArctiC Amplification: Climate Relevant Atmospheric and SurfaCe Processes, and Feedback Mechanisms  $(\mathcal{AC})^3$  in sub-project A02 "Balloon-borne observations and dedicated simulations of the transitions between typical states of the Arctic atmospheric boundary layer". The authors thank for providing them access to the infrastructure at Station Nord to perform the measurements. Thanks to Johannes Röttenbacher for his support during data publication and additional manuscript comments, as well as to Andreas Walbröl.

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
