# Peer review of "Tethered balloon-borne measurements to characterise the evolution of the Arctic atmospheric boundary layer at Station Nord"

_Earth System Science Data, 2025_

## Referee Comment (RC1)

Review of: **Tethered balloon-borne measurements to characterise the evolution of the Arctic atmospheric boundary layer at Station Nord**

Authors: Henning Dorff, Holger Siebert, Komal Navale, André Ehrlich, Joshua Müller, Michael Schäfer, Fan Wu, and Manfred Wendisch

**General Comments:**
This is a very well-written article describing an interesting balloon-based dataset. The authors have done a very nice job in describing the measurements and calculations in detail. I have few comments and think that the paper is more or less ready for publication. Below are a few relatively minor things for the authors to think about.

**Specific Comments:**
Abstract, Line 14: What are "temperature rates"? Should this be "Temperature advection rates"? "Sensible Heat Flux"? Something else?
Line 112: In calculating the winds, how do you handle the ground-relative velocity of the probe? Are we assuming this is zero? There are clearly times where the tether is moving (and hence, the probe is as well).
Line 136: "Deca-minute" is not a commonly used term. While people can figure it out, it will take most some time to think about what this means. Why not make it easier on the reader?
Line 153: Were the measurements time-stamped using a common clock on the data-logger? How was this done?
Line 184: Based on experience, even much smaller tilt angles have significant impacts. These impacts can be calculated and/or potentially corrected for. Has there been any attempt to do so?
Line 216: I do see the tail of the spectrum starting to flatten out. It might be helpful to include an average value as well, given that the spectrum itself has a relatively large range.
Line 278 and throughout manuscript: relative humidity? specific humidity? Please state clearly what is meant by "humidity" throughout the text.
~Lines 335-350: This is starting to border on analysis and discussion on model performance, which is typically not what ESSD articles are for. Recommend removing, as this is only a very limited evaluation. I appreciate that it is meant as an example, but I'm not convinced this is necessary.

**Technical Corrections:**
I didn't find any, which is very rare! I wonder whether the authors deployed an LLM to help ensure that there aren't any spelling, grammatical, or other issues? If so, great!

---

## Author Comment (AC1)

**Response to the comments from Anonymous Referee 1 for the submitted ESSD paper; Dorff, H. et al. (2025): "Tethered balloon-borne measurements to characterise the evolution of the Arctic atmospheric boundary layer at Station Nord"**

**Prefaces:**

We thank the ESSD handling editor, Fan Mei, as well as the Anonymous Referee #1, for the enlightening review. Please find our responses (in standard font) to the remarks from the Anonymous Referee #1 (in *italics*) below. Modifications in the manuscript are **bold.**

**General note:** To better acknowledge and account for the Villum Research Station, which was the basic measurement location in Station Nord for the BELUGA profiles, we now refer to "Villum Research Station" explicitly, and thus have changed the abbreviation for the measurement station location from "STN" to "VRS" throughout the manuscript.

**Responses to Reviewer 1:**

*This is a very well-written article describing an interesting balloon-based dataset. The authors have done a very nice job in describing the measurements and calculations in detail. I have few comments and think that the paper is more or less ready for publication. Below are a few relatively minor things for the authors to think about.*

**Response:** First of all, we want to sincerely thank you for your kind and helpful feedback, as well as for appreciating our work. We are confident that addressing your remarks will further improve our manuscript.

**Specific comments:**

*Abstract, Line 14: What are "temperature rates"? Should this be "Temperature advection rates"? "Sensible Heat Flux"? Something else?*
**Response:** We agree that the term is unprecise. We have changed it to: **"temporal temperature changes"**.

*Line 112: In calculating the winds, how do you handle the ground-relative velocity of the probe? Are we assuming this is zero? There are clearly times where the tether is moving (and hence, the probe is as well).*
**Response:** Indeed, there is some drifting of the tether and probe, but we assume this to be of minor relevance. Based on another balloon-borne Arctic campaign at Ny-Ålesund (Svalbard), we have calculated the oscillations of the tether (Fig. R1). The drift speeds, with a mean drift speed of 0.6 m/s, are about one order of magnitude smaller than the wind speeds. Therefore, we neglect them for simplicity.
**Modifications (L207ff):** Within the manuscript, we now explicitly mention this simplification in a later section (when further describing the measurement data handling and processing, Sect. 3.2) as follows: **"Furthermore, changes of the tether inclination and a related drift of the instrument probes can affect the measured wind speeds. Past BELUGA campaigns**

**have shown that the drift speeds of the probes are up to about one order of magnitude smaller than wind speeds. Therefore, these effects have been neglected in the data processing.”**

[Figure]

*Figure R1: Comparison of measured wind speeds and calculated drift speeds of the balloon-borne instrument probes for profile measurements conducted under Arctic conditions at Svalbard in 2024. For this data, mean drift speed was about 0.6 m/s per ascent.. For data users aiming at wind measurements with high precision, a correction of these effects by using the GPS data is recommended.*

*Line 136: “Deca-minute” is not a commonly used term. While people can figure it out, it will take most some time to think about what this means. Why not make it easier on the reader?*

**Response:** We reformulated as: **“at frequencies of ~10 min.”**

*Line 153: Were the measurements time-stamped using a common clock on the data-logger? How was this done?*

**Response:** For all the probes, we used GPS as the absolute time server, which was sufficient for 1 Hz data.

**Modifications (L156): “[...] referencing the UTC time (from probe-included GPS modules)”**

*Line 184: Based on experience, even much smaller tilt angles have significant impacts. These impacts can be calculated and/or potentially corrected for. Has there been any attempt to do so?*

**Response:** We agree that even smaller tilt angles can influence the measured irradiance. However, in contrast to solar radiation, the thermal infrared irradiance is nearly isotropic and, therefore, much less affected by the instrument tilt (Bucholtz et al., 2008: https://doi.org/10.1175/2008JTECHA1085.1). According to Wendisch and Yang (2012, 10.1080/00107514.2012.732967), the downward and upward irradiance are defined as hemispheric integrals:

$$F \downarrow = \int_0^{2\Pi} \int_{\Pi/2}^{\Pi} I \downarrow \cdot cos\theta \cdot sin\theta \; d\theta d\varphi,$$

$$F \uparrow = \int_0^{2\Pi} \int_0^{\Pi/2} I \uparrow \cdot cos\theta \cdot sin\theta \; d\theta d\varphi,$$

with the downward and upward broadband radiances *I.* For a Lambertian radiator, a tilted sensor receives a contribution from both hemispheres as illustrated in Fig. R2. There arises

contributions from both major directional components (upward and downward). As described in Picard et al. (2020), e.g., the measured downward irradiance ($F \downarrow$) is modified by a fraction of radiation from the opposite hemisphere proportional to $cos(\beta)$, following:

$F \downarrow (\beta) = (1 - cos(\beta)) \cdot F \uparrow + cos(\beta) F \downarrow$.

Assuming similar upward and downward irradiance values and an instantaneous sensor response as a worst-case scenario, the resulting uncertainty in the irradiance values increases with the pitch angle $\beta$. This leads to an uncertainty of approximately 0.4 % for $\beta$=5° and, nearly 4 % for $\beta$=15°. Nonetheless, since our data do not include solar radiation with much stronger directional dependency, we argue that the thresholds we used for the irradiance data quality flags are still appropriate, given the uncertainties above.

No tilt corrections were applied to our data because we aim to allow users full flexibility for individual post-processing in this regard. Following Picard et al. (2020; https://doi.org/10.5194/tc-14-1497-2020), the irradiances can be further corrected by accounting for the above-described cosine-dependent irradiance contribution caused by the sensor tilt and by considering the response time of the sensor.

**Modifications (L195ff):** We suggest adding the following sentence: **"[...] with uncertainties in the irradiance values greater than 5 %. This amounts to less than 5% of the total data and often are for measurements only shortly after launch. For users interested in considering this strong tilted data, correction routines as described by Picard et al. (2020) are suggested."**

[Figure]

*Fig. R2: Geometry of a sloped infinite plane exhibited by radiation.*

*Line 216: I do see the tail of the spectrum starting to flatten out. It might be helpful to include an average value as well, given that the spectrum itself has a relatively large range.*

**Response:** This has been addressed by using the smoothed spectrum, indicated in Fig. 7 (dark-green, dashed). This averaged spectrum is based on logarithmically equivalent bins. However, within the original manuscript, the figure caption referred to incorrect colours, which made it difficult to identify the lines; this has now been corrected (Response to Referee 2). In

addition, for greater clarity, we renamed the legend entry from "smoothed spectrum" to **"averaged spectrum"** in Fig. 7. A slight flattening is indeed visible at the highest frequencies, which indicates where sensor white noise from the sensor becomes dominant. However, the averaged spectrum still follows a straight line with a slope comparable to the inertia subrange model. We therefore argue that fluctuations up to more than 10 Hz remain robust.

**Modification (L231ff):** We specified that we meant the averaged spectrum: **"Up to frequencies slightly above 10 Hz, the averaged spectrum in Fig. 7 has not yet transitioned to white noise (a flat spectrum), implying that spatial structures of about 2 cm can still be well resolved."**

*Line 278 and throughout manuscript: relative humidity? specific humidity? Please state clearly what is meant by "humidity" throughout the text.*

**Response:** We apologize for the imprecise wording and have now ensured a clear and consistent specification of the moisture quantities throughout the manuscript. Only in selected cases, where referring generally to humidity is sufficient, we retained the generic term, as all moisture quantities can be derived and analysed individually from the measurements.

*Lines 335-350: This is starting to border on analysis and discussion on model performance, which is typically not what ESSD articles are for. Recommend removing, as this is only a very limited evaluation. I appreciate that it is meant as an example, but I'm not convinced this is necessary*

**Response:** We apologize for moving too far beyond the intended scope of ESSD with our exemplary case description. Our intention was to highlight the potential of the data for further analysis and encourage users to use the data especially for model and reanalysis evaluation by these appetizers. CARRA is one of the highest resolution state-of-the-art Arctic reanalysis, which plays an important role in the Arctic research community. For this reason, we considered it relevant to showcase the potential of the dataset for future studies.

However, we agree that when beginning to analyse thermodynamic profiles of CARRA and BELUGA observations, this could be interpreted as a discussion of model performance, and that such an assessment would be too limited and not sufficiently robust within the scope of this manuscript, which must remain focused on the data itself, as required by ESSD.

Therefore, when introducing Section 5 dealing with the general potential of the BELUGA observations to study Arctic ABL transition events, we motivated the importance of combining the observations with model data as follows:

**Modification (L301ff): "The following section sketches how BELUGA L2 data from VRS during such transitions can enhance our understanding of Arctic ABL steering processes, for example, regarding temporal temperature changes and radiative heating rates under varying cloud conditions. To fully exploit the BELUGA data and draw robust conclusions about the observed processes, future work will also rely on joint analyses with numerical weather prediction models or reanalysis data. An important aspect in this context is the comparability of the BELUGA data with the model data, particularly regarding the vertical resolution, which we introduce here for one of the LLJ events and the reanalysis data."**

Thus, we did not remove Section 5.3 and Figure 14, but we substantially revised the text. By this, the section now focuses on sketching potential future comparison studies and emphasizes that the general agreement between our measurements primarily serves to place our observations in a broader context. We highlight the value of the dataset for revealing

limitations that reanalyses exhibit within the Arctic ABL, using CARRA as an illustrative example.

**Modification (L355ff):** To reflect this clearer focus, we renamed the section to **"Low-level jet evolution"**, as this is an important phenomenon in the Arctic ABL that can be studied with our data. The revised section reads as follows: **"LLJs play a significant role in modulating the structure of the Arctic ABL structure due to their influence on vertical entrainment. Balloon-borne observations are well suited for studying LLJ events. Previous balloon-borne studies have demonstrated that LLJs can effectively enhance near-surface horizontal transport of passive tracers (Egerer et al., 2023).**

**During the measurement campaign at VRS, BELUGA sampled several LLJ events, as listed in Tab. 2 and shown in Fig. 11. A prominent LLJ example is presented in Fig. 14, illustrating vertical profiles of potential temperature, wind speed and specific humidity from RF12 on 1 April 2024. The BELUGA observations reveal a pronounced LLJ with a maximum wind speed approaching 5 m s$^{-1}$ above a stably stratified Arctic ABL extending up to about 100 m (Fig. 14a, b). Over the course of the flight, the LLJ intensity increased by approximately 1 m/s (Fig. 14b). Apart from a decrease in the near-surface layer (<50 m), moisture conditions remained fairly homogeneous throughout the lowest 500 m with $q$ between 0.4 and 0.5 g kg$^{-1}$ (Fig. 14c).**

**These measurements allow the structure of LLJs in Arctic ABLs to be compared with reanalysis profiles. As an example, Fig. 14 includes the corresponding CARRA reanalysis profiles for the duration of the flight. The overall agreement between BELUGA and CARRA, particularly regarding potential temperature (Fig. 14a) and the general shape and altitude of the LLJ, places the observations in a broader meteorological context. The high-resolution vertical profiles of meteorological, radiative, and turbulence conditions obtained with BELUGA provide valuable case-study material for future evaluations of high-resolution reanalyses such as those provided by CARRA. The observations support efforts to better understand limitations in the model-based representation of transition events. Differences in specific humidity and maximum wind speed at the LLJ height for this case in CARRA (Fig. 14b, c) illustrate aspects that could be examined in more detail across additional LLJ events."**